EMBO
Molecular Medicine

# *DOK7* gene therapy enhances motor activity and life span in ALS model mice

Sadanori Miyoshi[1] (ID), Tohru Tezuka[1], Sumimasa Arimura[1], Taro Tomono[2,3], Takashi Okada[2] & Yuji Yamanashi[1,*] (ID)

## Abstract

**Amyotrophic lateral sclerosis (ALS) is a progressive, multifactorial motor neurodegenerative disease with severe muscle atrophy. The glutamate release inhibitor riluzole is the only medication approved by the FDA, and prolongs patient life span by a few months, testifying to a strong need for new treatment strategies. In ALS, motor neuron degeneration first becomes evident at the motor nerve terminals in neuromuscular junctions (NMJs), the cholinergic synapse between motor neuron and skeletal muscle; degeneration then progresses proximally, implicating the NMJ as a therapeutic target. We previously demonstrated that activation of muscle-specific kinase MuSK by the cytoplasmic protein Dok-7 is essential for NMJ formation, and forced expression of Dok-7 in muscle activates MuSK and enlarges NMJs. Here, we show that therapeutic administration of an adeno-associated virus vector encoding the human *DOK7* gene suppressed motor nerve terminal degeneration at NMJs together with muscle atrophy in the *SOD1-G93A* ALS mouse model. Ultimately, we show that *DOK7* gene therapy enhanced motor activity and life span in ALS model mice.**

**Keywords** amyotrophic lateral sclerosis; *DOK7*; gene therapy; neuromuscular junction

**Subject Categories** Genetics, Gene Therapy & Genetic Disease; Neuroscience

## Introduction

Amyotrophic lateral sclerosis (ALS) is a fatal neuromuscular disease with motor neuron degeneration that causes muscle weakness, paralysis, and respiratory failure (Paez-Colasante *et al*, 2015). It has been established that ALS is a multifactorial disease, and indeed, many abnormalities have been identified as potential pathogenic factors in patients with ALS and its model animals, including accumulation of protein aggregates, defective RNA processing, oxidative stress, glutamate excitotoxicity, glial dysfunction, and abnormal muscle energy

metabolism (Paez-Colasante *et al*, 2015; Zhu *et al*, 2015). This variation appears to be a roadblock to development of effective therapy (Genç & Özdinler, 2014; Ittner *et al*, 2015). Currently, the glutamate release inhibitor riluzole is the only medication approved by the FDA for ALS, and prolongs patient life span by a few months (Lucette *et al*, 1996). Although many other drugs have been under clinical trial, there remains a strong need for new treatment strategies for ALS (Ittner *et al*, 2015). Recent studies of ALS model mice revealed that motor neuron degeneration first becomes evident as size reduction of the motor nerve terminals and subsequent denervation at neuromuscular junctions (NMJs), a cholinergic synapse essential for motoneural control of muscle contraction (Fischer *et al*, 2004; Murray *et al*, 2010; Dadon-Nachum *et al*, 2011; Valdez *et al*, 2012). Motor neuron degeneration then progresses proximally. This pattern, known as "dying-back" pathology or distal axonopathy, is also observed in autopsy or electrophysiology of ALS patients (Fischer *et al*, 2004; Dadon-Nachum *et al*, 2011; Bruneteau *et al*, 2015), suggesting that NMJ protection might be an effective treatment.

In mammals, the formation and maintenance of NMJs are orchestrated by the muscle-specific receptor tyrosine kinase MuSK (Burden, 2002), which requires Dok-7 as an essential, muscle-intrinsic activator (Okada *et al*, 2006; Inoue *et al*, 2009). Indeed, recessive mutations in the human *DOK7* gene cause the congenital myasthenic syndrome *DOK7* myasthenia, which is characterized by defective NMJ structure or NMJ synaptopathy (Beeson *et al*, 2006). Previously, we generated AAV-D7, a recombinant muscle-tropic adeno-associated virus (AAV) serotype 9 vector carrying the human *DOK7* gene tagged with EGFP under the control of the cytomegalovirus promoter, and demonstrated that therapeutic administration of AAV-D7 enlarges NMJs and enhances motor activity and life span in *DOK7* myasthenia model mice (Arimura *et al*, 2014). Furthermore, therapeutic administration of AAV-D7—*DOK7* gene therapy—also enlarged NMJs and enhanced motor activity and life span in a mouse model of autosomal dominant Emery–Dreifuss muscular dystrophy, a disease associated with defective NMJs due to mutations in the lamin A/C gene (Méjat *et al*, 2009). Although these observations demonstrate potential for *DOK7* gene therapy in these myopathies with NMJ defects, we suspected that this therapy might also benefit motor neurodegenerative diseases, because muscle-specific

1   Division of Genetics, The Institute of Medical Science, The University of Tokyo, Tokyo, Japan
2   Department of Biochemistry and Molecular Biology, Nippon Medical School, Tokyo, Japan
3   Graduate School of Comprehensive Human Sciences, Majors in Medical Sciences, University of Tsukuba, Ibaraki, Japan
    *Corresponding author. Tel: +81 3 6409 2115; Fax: +81 3 6409 2116; E-mail: yyamanas@ims.u-tokyo.ac.jp

overexpression of Dok-7 activates MuSK and enlarges not only the postsynaptic apparatus but also presynaptic motor nerve terminals at NMJs, which might counteract degeneration, or size reduction in particular, of the terminal (Inoue *et al*, 2009; Arimura *et al*, 2014). Since degeneration of motor nerve terminals appears to play an important role in pathogenesis of both patients and animal models of ALS (Fischer *et al*, 2004; Murray *et al*, 2010; Dadon-Nachum *et al*, 2011; Valdez *et al*, 2012; Bruneteau *et al*, 2015), in the present study we examined whether *DOK7* gene therapy ameliorates pathology in a mouse model of ALS.

# Results

## *DOK7* gene therapy suppresses motor nerve terminal degeneration at the NMJ in ALS mice

In ~20% of familial cases of ALS, patients harbor a gain-of-function mutation in the gene encoding Cu/Zn superoxide dismutase 1 (SOD1; Ittner *et al*, 2015; Paez-Colasante *et al*, 2015). Moreover, C57BL/6 mice expressing human SOD1 (hSOD1) with the ALS-linked G93A mutation (hereafter ALS mice) manifest progressive muscle paralysis similar to that observed in clinical cases along with the histopathological hallmarks observed in familial and sporadic ALS, including NMJ defects (Dadon-Nachum *et al*, 2011; Zhu *et al*, 2015). Based on these pathological features, ALS mice have been widely employed as a useful model, although following the guideline for preclinical ALS studies is strongly recommended for achieving results that better translate into humans (see below; Ludolph *et al*, 2010). Thus, in accord with the guideline, we first intravenously administered $1.2 \times 10^{12}$ viral genomes (vg) of AAV-D7 to male ALS mice that as a group were at an early symptomatic stage as reported elsewhere (postnatal day 90, P90; Kondo *et al*, 2014), and performed histological analyses 30 days later. Note that the forelimb grip strength of the control and AAV-D7 treatment groups of ALS mice was significantly weaker than wild-type (WT) mice at P84 (before treatment), confirming that these ALS mice, as a group, were already at an early symptomatic stage (Fig 1A). Also, we confirmed that the *hSOD1-G93A* transgene copy number, substantial variation in which might affect disease progression (Pérez-García & Burden, 2012), was comparable in both groups of ALS mice (Fig 1B). In these model mice, AAV-D7 treatment robustly enhanced MuSK activation, as judged by phosphorylation of MuSK and acetylcholine receptor (AChR) in the hind-limb muscle, the latter known to be triggered by activation of MuSK (Fig 1C). To examine histopathology, we first studied NMJs in the diaphragm muscle, where NMJs are particularly amenable to whole-mount imaging due to the muscle's thin and planar structure (Tetruashvily *et al*, 2016) and where changes in neuromuscular transmission occur long before motor symptom onset (Rocha *et al*, 2013). In non-treated ALS mice, although the postsynaptic area characterized by clustered AChRs was not affected, the area of motor nerve terminals was significantly decreased, supporting the neuropathic nature of the defects (Fig 1D–F). However, AAV-D7 treatment significantly increased the area of motor nerve terminals at NMJs, together with the postsynaptic area, indicating positive effects on motor neurons. Indeed, denervation at NMJs was significantly suppressed by the treatment (Fig 1G), although these NMJs showed partial

innervation, partly due to the primary enlargement of the postsynaptic area (Fig 1D–F). Together, these findings demonstrate that *DOK7* gene therapy protected NMJs from nerve terminal degeneration in ALS mice. Since AAV-D7 expresses human Dok-7 tagged with EGFP, its infection could be monitored (Figs 1D and EV1A–C).

## *DOK7* gene therapy suppresses muscle atrophy with no adverse effects on proximal motor neuron degeneration in ALS mice

Because NMJ defects are thought to be a cause of muscle atrophy in ALS (Pansarasa *et al*, 2014), we further evaluated effects of AAV-D7 treatment on muscle fiber size. Analysis of size distribution of the fiber cross-sectional areas (CSA) in tibialis anterior muscle showed that AAV-D7 treatment apparently increased the proportion of myofibers with relatively larger diameter in ALS mice compared with non-treated ALS mice (Fig 2A–C). Consistently, both the cumulative percentage and mean value of CSA showed significant improvement, indicating a beneficial effect of AAV-D7 treatment on muscle atrophy in ALS mice (Fig 2D and E). We also examined the effect of AAV-D7 treatment on progressive motor neuron death, another hallmark of ALS (Yoo & Ko, 2012), and found no significant changes: The number of motor neurons in the ventral horns of the L4-L5 spinal cord segments was comparable between AAV-D7-treated and non-treated ALS mice (Fig EV2A and B). Similarly, proximal axon atrophy was not significantly changed, as the caliber distribution of L4 ventral root axons was comparable between the two groups (Fig EV2C and D). Together, our data indicate that *DOK7* gene therapy suppressed NMJ defects and muscle atrophy in ALS mice with no obvious effects on proximal motor neuron degeneration at 30 days postinfection.

## *DOK7* gene therapy prolongs life span in ALS mice

Since an individual patient with ALS is usually diagnosed after onset of symptoms (Zhu *et al*, 2015), we defined disease onset of each ALS mouse individually as follows, in order to examine the effects of *DOK7* gene therapy on life span and motor activity. First, we set the reference points at P84 and P86, within an early symptomatic stage as a group (Fig 1A), to individually obtain a reference forelimb grip strength for each ALS mouse as the mean of those at P84 and P86. Then, disease onset for each ALS mouse was defined as when its grip strength dropped to 80% or less of its own reference value for two consecutive days, since ALS mice sometimes show ~10% fluctuation even over 2 days. At this individually defined disease onset, we intravenously administered $1.2 \times 10^{12}$ vg of AAV-D7 into each given male ALS mouse to perform survival analysis according to the guideline for preclinical ALS studies (Ludolph *et al*, 2010), which requires a minimum of 12 mice of a single gender and of comparable *hSOD1-G93A* transgene copy number for each test or control group. Note that this guideline was recently established for better translation into humans, mainly because many, if not all, clinical trials had failed to translate beneficial life span effects in ALS mice into humans (Genç & Özdinler, 2014). Remarkably, a study using experimental settings that fit this guideline recently reported that even the FDA-approved riluzole failed to prove its efficacy in ALS mice (Jablonski *et al*, 2014). However, *DOK7* gene therapy significantly prolonged life span of ALS mice ($166.3 \pm 3.2$ days) compared with non-treated ALS mice ($154.4 \pm 2.7$ days; Fig 3A). This therapy also

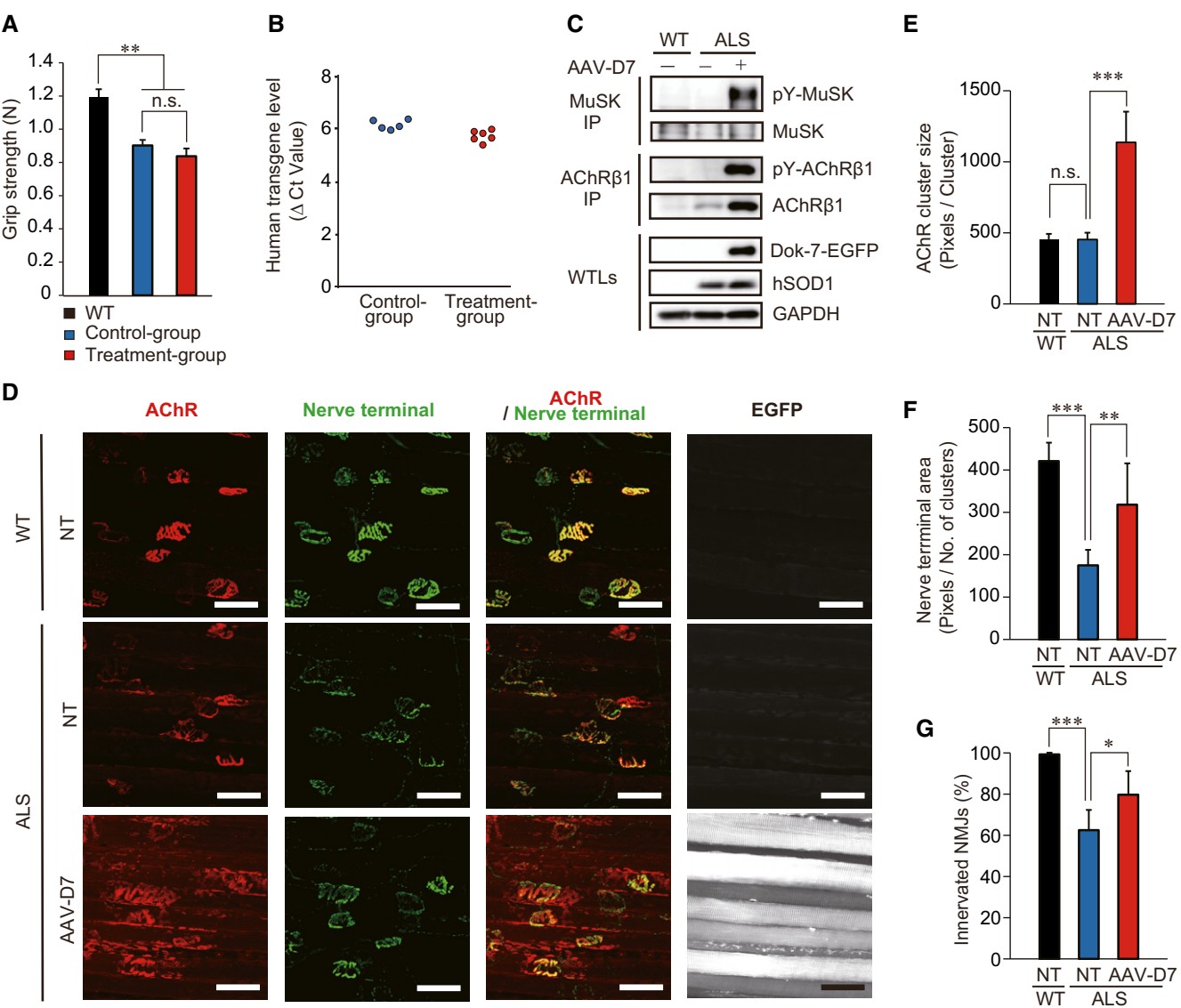

**Figure 1. *DOK7* gene therapy suppresses motor nerve terminal degeneration at the NMJ in ALS mice.**

A Forelimb grip strength of wild-type (WT) mice and that of control and AAV-D7 treatment groups of ALS mice at P84 (WT, *n* = 5 mice; control group, *n* = 5 mice; treatment group, *n* = 6 mice). Values are means ± SEM. **P = 0.0016 (WT vs. control group) and 0.0003 (WT vs. treatment group) by one-way analysis of variance (ANOVA) with Bonferroni's *post hoc* test. n.s., not significant.

B The difference in cycle threshold ($\Delta C_t$) between the human *SOD1-G93A* transgene and the reference mouse *apob* gene. To calculate the human transgene level, the $\Delta C_t$ value of *hSOD1* was subtracted from the $\Delta C_t$ value of *apob* (control group, *n* = 5 mice; treatment group, *n* = 6 mice).

C–G WT or ALS mice treated or not with $1.2 \times 10^{12}$ vg of AAV-D7 at P90 were subjected to the following assays at P120. Tyrosine phosphorylation of MuSK or AChR in the hind-limb muscle. MuSK or AChRβ1 subunit (AChRβ1) immunoprecipitates (IP) from whole-tissue lysates (WTLs) of the hind-limb muscle were immunoblotted for phosphotyrosine (pY), MuSK, and AChRβ1. WTLs were blotted for Dok-7-EGFP, human SOD1, and GAPDH (C). Whole-mount staining of NMJs on the diaphragm muscle. Motor nerve terminals (green) and postsynaptic AChRs (red) were stained with anti-synapsin-1 antibodies and α-bungarotoxin, respectively. Expression of Dok-7-EGFP fusion protein (gray) was monitored by EGFP. Scale bars, 50 μm. NT, not treated (D). The size of AChR cluster (E), the size of motor nerve terminal (F), and innervation ratio (G) (WT-NT, *n* = 5 mice; ALS-NT, *n* = 5 mice; ALS-AAV-D7, *n* = 6 mice). Values are means ± SD. (E) ***P < 0.0001; (F) **P = 0.0077, ***P = 0.0001; (G) *P = 0.0134, ***P < 0.0001 by one-way ANOVA with Dunnett's *post hoc* test.

Source data are available online for this figure.

prolonged duration of survival after ALS onset (64.2 ± 3.3 days) compared with non-treated mice (50.3 ± 3.1 days; Fig 3B). Neither the age nor the grip strength at disease onset was statistically different between AAV-D7-treated and non-treated groups (Fig 3C and D). In addition, the *hSOD1-G93A* transgene copy number was comparable between these groups (Fig 3E). We also confirmed that

treatment with AAV-EGFP did not significantly alter either the survival (162.1 ± 2.5 days) nor duration after onset (59.8 ± 3.0 days) compared with non-treated mice (survival: 158.4 ± 1.9 days, duration after onset: 57.3 ± 2.4 days), showing that neither AAV infection nor EGFP expression had any obvious effect on survival (Fig EV3A–E). Note that *DOK7* gene therapy did

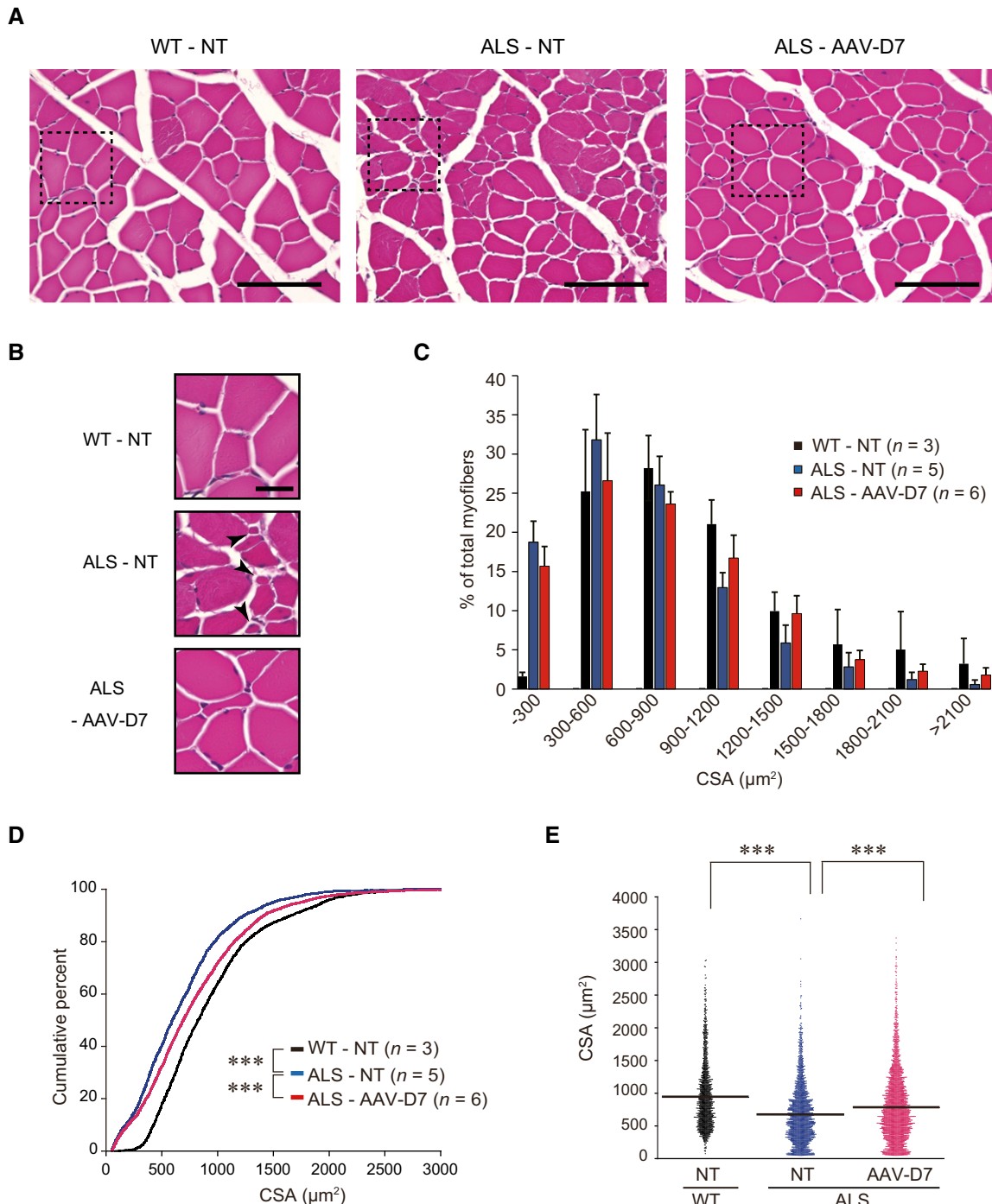

**Figure 2. *DOK7* gene therapy suppresses myofiber atrophy in ALS mice.**
WT or ALS mice treated or not with $1.2 \times 10^{12}$ vg of AAV-D7 at P90 were analyzed at P120.

A   Hematoxylin and eosin (H&E) staining of transverse sections of tibialis anterior muscle. Scale bars, 100 μm. NT, not treated.

B   Magnified views of boxed regions in (A). Arrowheads indicate severe myofiber atrophies in ALS-NT. Scale bar, 25 μm.

C   Size distribution of the myofiber cross-sectional areas (CSA). Values are means ± SEM.

D   Cumulative percentage of myofiber CSA. ***$P < 0.0001$ by Kolmogorov–Smirnov test.

E   Individual and mean CSA (WT-NT, $n = 3$ mice; ALS-NT, $n = 5$ mice; ALS-AAV-D7, $n = 6$ mice). ***$P < 0.0001$ by one-way ANOVA with Dunnett's *post hoc* test.

not affect degeneration of motor neuron cell bodies even at end stage (P150) in ALS mice (Fig EV4A and B), implying that the protective effect of this therapy may be confined distally at the NMJ. This is apparently consistent with the reports that degeneration of motor nerve terminals at NMJs occurs independently of motor neuron cell death (Kostic *et al*, 1997; Gould *et al*, 2006; Parone *et al*, 2013). In

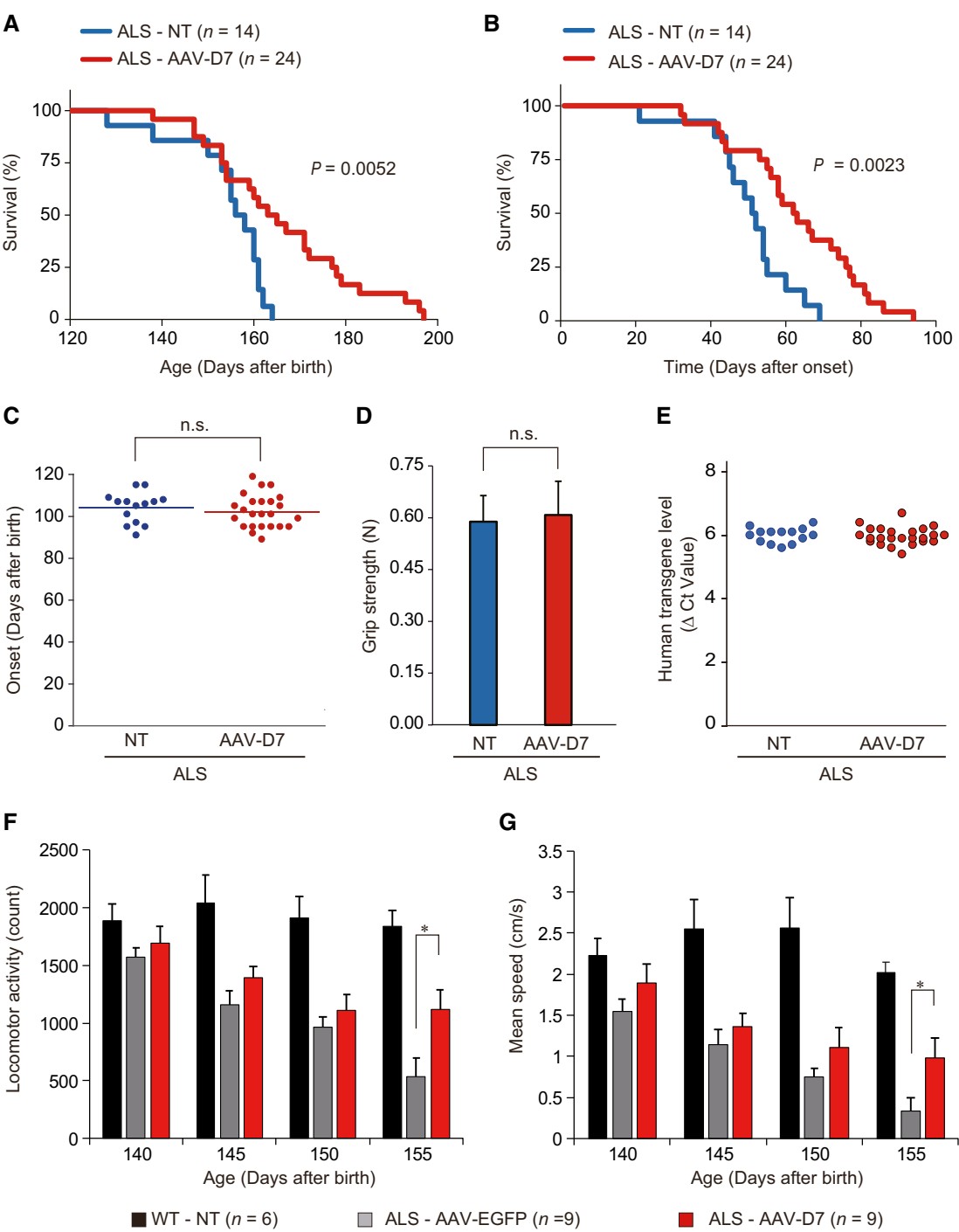

**Figure 3.  *DOK7* gene therapy enhances life span and motor activity in ALS mice.**

A–E   ALS mice treated or not with 1.2 × 10$^{12}$ vg of AAV-D7 at individually defined disease onset were subjected to the following assays. Kaplan–Meier survival curves after birth of untreated ALS mice ($n$ = 14 mice, 154.4 ± 2.7 days, mean ± SEM) and AAV-D7-treated ALS mice ($n$ = 24 mice, 166.3 ± 3.2 days). $P$ = 0.0052 by log-rank test. NT, not treated (A). Kaplan–Meier survival curves after onset in untreated ALS mice ($n$ = 14 mice, 50.3 ± 3.1 days, mean ± SEM) and AAV-D7-treated ALS mice ($n$ = 24 mice, 64.2 ± 3.3 days). $P$ = 0.0023 by log-rank test (B). Ages at onset for untreated ($n$ = 14 mice) and AAV-D7-treated ALS mice ($n$ = 24 mice). Mean ages of onset were indicated by horizontal bars. Values are means ± SEM. n.s., not significant by Student's $t$-test (C). Forelimb grip strength of ALS mice at onset in untreated ($n$ = 14 mice) and AAV-D7-treated groups ($n$ = 24 mice). Values are means ± SEM. n.s., not significant by Student's $t$-test (D). The difference in cycle threshold ($\Delta C_t$) between the human *SOD1-G93A* transgene and the reference mouse *apob* gene. To calculate the human transgene level, the $\Delta C_t$ value of hSOD1 was subtracted from the $\Delta C_t$ value of apob (ALS-NT, $n$ = 14 mice; ALS-AAV-D7, $n$ = 24 mice) (E).

F, G   WT or ALS mice treated with 1.2 × 10$^{12}$ vg of AAV-EGFP or AAV-D7 at individually defined disease onset were subjected to the open field tests at the indicated ages ($n$ = 6 or 9 mice). Spontaneous motor activity was represented by locomotor activity (F) and mean speed (G). Values are means ± SEM. (F) *$P$ = 0.0244, (G) *$P$ = 0.0349 by Mann–Whitney $U$-test.

addition, although we mentioned above that *DOK7* gene therapy might counteract size reduction of the motor nerve terminals at NMJs in ALS mice, there remains the possibility that this therapy might also facilitate reinnervation by remaining motor neurons. Thus, it is important to investigate how *DOK7* gene therapy inhibits degeneration of the motor nerve terminals in ALS mice.

### *DOK7* gene therapy improves motor activity in ALS mice

Although we monitored the grip strength of individual mice up to 152 days of age, the data fail to show a significant difference between AAV-D7-treated and non-treated ALS mice (Fig EV4C). However, as AAV-EGFP-treated ALS mice entered late- to end-stage disease, the contrast in spontaneous motor activity with AAV-D7-treated mice was striking (Movie EV1). Thus, we used the automated home cage behavioral system to measure stress-independent spontaneous motor activity (Hayworth & Gonzalez-Lima, 2009; Ludolph *et al*, 2010). The data show an increase resulting from AAV-D7 treatment throughout the testing period and demonstrate that *DOK7* gene therapy significantly improved locomotor activity and speed compared with AAV-EGFP-treated mice at P155 (Fig 3F and G).

## Discussion

Accumulating evidence demonstrates that defects in NMJs are associated with various types of neuromuscular disorders, including myasthenia, motor neuron degeneration, and sarcopenia, an age-related muscle atrophy (Deschenes *et al*, 2010; Murray *et al*, 2010; Sleigh *et al*, 2014). For example, patients with ALS, including one carrying a *SOD1* mutation, manifested NMJ defects, even when biopsied at the early symptomatic stage (Bruneteau *et al*, 2015). Consistent with this, mouse models of ALS with *SOD1, TDP43, FUS*, or *C9ORF72* mutations also display NMJ defects (Dadon-Nachum *et al*, 2011; Arnold *et al*, 2013; Liu *et al*, 2016; Sharma *et al*, 2016). In addition, mouse models of neurodegenerative diseases including spinal muscular atrophy and Charcot–Marie–Tooth disease type 2D show NMJ abnormalities (Murray *et al*, 2010; Sleigh *et al*, 2014). Furthermore, denervation at the NMJ occurs before myofiber atrophy in sarcopenia model rats (Deschenes *et al*, 2010), highlighting the NMJ as an attractive therapeutic target. As mentioned, we previously demonstrated that *DOK7* gene therapy activates muscle-specific kinase MuSK to enlarge NMJs, and ameliorates myopathies characterized by NMJ defects in mouse models irrespective of the presence or absence of a *dok-7* gene mutation, attesting to its potential in a range of neuromuscular diseases (Arimura *et al*, 2014). However, *DOK7* gene therapy has not previously been demonstrated to be effective in motor neuron diseases. Here we demonstrated that a single-dose treatment of AAV-D7 after individually defined disease onset improved life span and motor activity in ALS mice, although the mechanisms by which *DOK7* gene therapy suppresses denervation at the NMJ, and disease progression in ALS, remain to be established. Given that EGFP fluorescence from AAV-D7-infected cells was detectable however weakly and faintly in the spinal cord and cerebellum, respectively, in AAV-D7-treated mice (Fig EV1B and C), skeletal muscle-specific transduction of *DOK7* will be necessary to completely exclude the possible contribution of non-muscle transgene expression. In addition, to clarify the therapeutic effects of

NMJ protection on muscle weakness, electrophysiological study of affected muscle will be essential.

A few studies have evaluated the effect of NMJ preservation on ALS. The expression level of Nogo-A, an inhibitor of neurite outgrowth, is elevated in ALS (Jokic *et al*, 2005), and its genetic depletion delays denervation at NMJs and prolongs life span in ALS model mice (Jokic *et al*, 2006). Consistent with this, treatment with anti-Nogo-A antibody improved muscle innervation and motor function in ALS model mice. However, its effect on survival was not shown (Bros-Facer *et al*, 2014). Moreover, therapeutic treatment with anti-Nogo-A antibody (Ozanezumab) did not show any benefits for ALS patients in a phase II clinical trial (Meininger *et al*, 2017). In addition, mislocalization of MuSK was reported in *SOD1-G93A* ALS model mice (Vilmont *et al*, 2016), and a transgene that modestly increases MuSK expression in muscle from the embryonic stage delayed denervation at NMJs and improved motor function but failed to increase survival of ALS mice (Pérez-García & Burden, 2012). Nevertheless, the effect of ectopic MuSK expression induced after disease onset is not known, and inborn higher-level expression of MuSK in the muscle has been shown to induce scattered NMJ formation throughout myofibers and cause severe muscle weakness (Kim & Burden, 2008), suggesting that forced MuSK expression is not suitable as a therapeutic method. By contrast, we previously showed that *DOK7* gene therapy greatly facilitates NMJ enlargement in the appropriate central region of myofibers without lethal effects for more than 1 year in *DOK7* myasthenia model mice (Arimura *et al*, 2014), suggesting that *DOK7* gene therapy is a safer therapeutic approach.

Our findings demonstrate that *DOK7* gene therapy has potential for treating various motor neuron diseases that manifest NMJ defects. Pharmacological enlargement of NMJs might also be useful. Because previous studies have shown that degeneration of motor nerve terminals at NMJs occurs independently of motor neuron cell death in ALS model mice as mentioned above (Kostic *et al*, 1997; Gould *et al*, 2006; Parone *et al*, 2013), these NMJ-targeted therapies might be more effective when used in combination with other therapies such as those aimed at promoting motor neuron survival.

## Materials and Methods

### Mice

The animal studies were performed in accordance with the University of Tokyo guidelines for animal care and use, and approved by the institutional animal care and use committee. Transgenic mice expressing human SOD1 (hSOD1) with the ALS-linked G93A mutation (B6. Cg-Tg[SOD1-G93A]1Gur/J, Stock No. 004435) were purchased from The Jackson Laboratory and bred on the C57BL/6J background, and male mice were used in this study. All mice used in this study were housed on a 12-h light/dark cycle in specific pathogen-free conditions with free access to water and standard mouse chows in the animal facility of the Institute of Medical Science, the University of Tokyo.

### Evaluation of *hSOD1-G93A* transgene copy number

Changes in the transgene copy number were estimated using real-time quantitative PCR by determining the difference in cycle

threshold ($\Delta C_t$) between the transgene (*hSOD1-G93A*) and a reference gene (mouse *apob*), according to the recommendation by The Jackson Laboratory. SYBR Premix Ex Taq II (Takara Bio) was used for real-time amplification of DNA. Specific primers for *hSOD1-G93A* and *apob* sequences were as follows (5′ to 3′): GGGAAGCTG TTGTCCCAAG (*hSOD1-G93A*, forward), CAAGGGGAGGTAAAAGA GAGC (*hSOD1-G93A*, reverse), CACGTGGGCTCCAGCATT (*apob*, forward), and TCACCAGTCATTTCTGCCTTTG (*apob*, reverse).

## AAV production

The cDNA encoding EGFP or human Dok-7 cDNA tagged with EGFP was cloned into pAAV-MCS (Agilent Technologies), which carries the cytomegalovirus promoter, to obtain pAAV-EGFP or pAAV-Dok-7-EGFP plasmid (Arimura *et al*, 2014). For production of AAV-EGFP or AAV-D7, HEK293T or HEK293EB cells were co-transfected with the AAV9 chimeric helper plasmid pRep2Cap9, the adenovirus helper plasmid pHelper (Agilent Technologies), and pAAV-EGFP or pAAV-Dok-7-EGFP in a HYPERFlask vessel (Corning) using polyethylenimine and cultured for 5 days (Matsushita *et al*, 2004; Lin *et al*, 2007). The AAV particles were purified by density-gradient ultracentrifugation (Tomono *et al*, 2016). The viral titers were determined by real-time quantitative PCR using AAVpro Titration Kit (Takara Bio) with specific primers for the *EGFP* sequence as follows (5′ to 3′): GTGAGCAAGGGCGAGGAG (forward) and GTGGTGCAG ATGAACTTCAGG (reverse).

## *In vivo* AAV injection

$1.2 \times 10^{12}$ vg of AAV-D7 or AAV-EGFP was intravenously injected by a single shot via the tail vein at a symptomatic stage (P90) or individually defined disease onset. Note that we used wild-type or ALS mice without any treatment, even sham injection, as the control "non-treated mice" or "WT-NT" or "ALS-NT".

## Immunoprecipitation and immunoblotting

Whole-tissue lysates (WTLs) were prepared from the hind-limb muscle with alkaline lysis buffer [50 mM Tris–HCl (pH 9.5), 1% sodium deoxycholate, protease inhibitors (Complete, Roche), and phosphatase inhibitors (PhosSTOP, Roche)]. For immunoprecipitation, WTLs were incubated with antibodies to MuSK (C-19 and N-19) or AChRβ1 (H-101) (1:100, Santa Cruz Biotechnology), followed by incubation with protein G-Sepharose (GE Healthcare). For immunoblotting, immunoprecipitates or WTLs were applied to SDS–PAGE, and transferred to a polyvinylidene fluoride (PVDF) microporous membrane (Merck Millipore). The membranes were probed with primary antibodies for phosphotyrosine (4G10; 1:5,000, Merck Millipore), Dok-7 (A-7), AChRβ1 (H-101) (1:2,000, Santa Cruz Biotechnology), MuSK (AF562; 1:2,000, R&D Systems), hSOD1 (#2770), or GAPDH (#2118; 1:2,000, Cell Signaling Technology), followed by incubation with secondary horseradish peroxidase-labeled antibodies anti-mouse IgG (1:10,000, GE Healthcare), anti-rabbit IgG (1:10,000, GE Healthcare), anti-goat IgG (1:10,000, Santa Cruz Biotechnology), or TrueBlot anti-rabbit IgG antibodies (1:2,000, Rockland). The blots were visualized using a LAS4000 imager with ECL Prime Western Blotting Detection Reagent (GE Healthcare). The experiment was repeated four times independently.

## Whole-mount staining of NMJs

Diaphragm muscles were fixed in 1% paraformaldehyde (PFA) in phosphate-buffered saline (PBS) overnight at 4°C and then rinsed with PBS. The muscles were permeabilized with 1% Triton X-100 in PBS, and incubated with anti-synapsin-1 (#5297) rabbit monoclonal antibodies (1:1,000, Cell Signaling Technology) followed by incubation with Alexa 647-conjugated anti-rabbit IgG (1:2,000, Thermo) and Alexa 594-conjugated α-bungarotoxin (1:2,000, Thermo). Confocal Z serial images were collected with an FV1000 Confocal Laser Scanning Microscope (Olympus) and collapsed into a single image. Images were captured with the same settings and exposure time in each experimental group for comparison. The size (area) and number of presynaptic motor nerve terminals and postsynaptic AChR clusters were quantified using cellSens Digital Imaging Software (Olympus). For quantification, seven microscopic fields with the 20× objective were chosen at random on the diaphragm muscle from each mouse, and 170–260 synaptic sites were analyzed per mouse. These experiments were conducted in a blinded fashion.

## Quantification of myofiber size

Tibialis anterior (TA) muscles were fixed in 4% PFA in PBS and embedded in paraffin wax. Transverse sections of TA muscle were prepared at 7 μm thickness and subjected to hematoxylin and eosin (H&E) staining. Bright-field images of muscle bundles were collected with a BioREVO fluorescent microscope (Keyence). Cross-sectional area of TA muscle fiber was measured by cellSens Digital Imaging Software. For quantification, at least 450 myofibers per mouse were analyzed. These experiments were conducted in a blinded fashion.

## Visualizing of Dok-7-EGFP expression *in vivo*

Mice were transcardially perfused with 4% PFA in PBS under deep isoflurane anesthesia. The brain, spinal cord, tibialis anterior muscle, extensor digitorum longus muscle, soleus muscle, and diaphragm muscle were excised, postfixed overnight in 4% PFA in PBS, and placed in 15% sucrose PBS solution overnight and then in 30% sucrose PBS solution overnight. Tissues were frozen in a 1:1 mixture of 30% sucrose PBS and Tissue-tek O.C.T. Compound (Sakura Finetek). Transverse (muscle or spinal cord) or sagittal (brain) sections were prepared at 20 μm thickness and mounted with VECTASHIELD Hard Set Mounting Medium with DAPI (H-1500, Vector Laboratories, Inc.). Images were collected with an FV1000 confocal laser scanning microscope, and this imaging was performed with the same settings and exposure time for comparison. Data are representative of at least three mice.

## Quantification of motor neuron number

Mice were transcardially perfused with 4% PFA in PBS under deep isoflurane anesthesia. The spinal cord lumbar region was excised, postfixed overnight in 4% PFA in PBS, and embedded in paraffin wax. Transverse sections of the L4-L5 spinal segment were prepared at 5 μm thickness and stained with cresyl violet (Nissl staining). Bright-field images of spinal cord sections were collected with a BioREVO fluorescent microscope. The number of motor neurons in the ventral horn was counted using cellSens Digital Imaging

Software, and at least 20 sections were counted per mouse. The following criteria were used to count motor neurons: (i) cells located in the ventral horn, and (ii) cells with a maximum diameter of 20 μm or more (Cai *et al*, 2015). These experiments were conducted in a blinded fashion. Note that we validated this method by comparing the motor neuron number in wild-type or ALS mice at P120 (Fig EV2B) with those reported elsewhere (Yoo & Ko, 2012).

### Quantification of motor axon diameter

L4 ventral roots were frozen in Tissue-tek O.C.T. Compound. To visualize motor axons, transverse sections of L4 ventral roots were prepared at 5 μm thickness and stained with anti-neurofilament-H rabbit polyclonal antibodies (AB1991; 1:2,000, Merck Millipore) followed by incubation with Alexa 594-conjugated anti-rabbit IgG (1:2,000, Thermo). Images were collected with a BioREVO fluorescent microscope, and this imaging was performed with the same settings and exposure time for comparison. Axonal diameters of L4 roots were measured by cellSens Digital Imaging Software. These experiments were conducted in a blinded fashion.

### Grip strength test and onset definition

Forelimb grip strength of each mouse was measured using a computerized electronic pull-strain gauge 1027DSM (Columbus Instruments) as described previously (Eguchi *et al*, 2016). Five measurements were taken per mouse, and the mean of these five measurements was used for statistical analysis. These experiments were conducted in a blinded fashion. To individually define disease onset, the reference forelimb grip strength of each ALS mouse was set as the mean of its strength values at P84 and P86; then, the onset was defined as when its grip strength dropped to 80% or less of its own reference strength for two consecutive days. We measured forelimb grip strength of each mouse every 2 days, unless two-consecutive-day measurements were required to define the onset.

### Open field test

Spontaneous motor activity was monitored in the IR Actimeter system (Panlab/Harvard Apparatus). For each measurement, a mouse was placed in the test cage (155 × 245 × 148 mm) for 5 min before recording in order to avoid any bias due to stress. Then, its movement was automatically recorded for 10 min by infrared capture. We analyzed the locomotor activity (counts) and mean speed (cm/s) using the Actitrack software (Panlab/Harvard Apparatus). These experiments were conducted in a blinded fashion.

### Statistical analysis

Data were analyzed using JMP Pro 12 (SAS Institute Inc.) or Easy R software. Values are presented as means ± SEM or ± SD. Statistical differences between two groups were determined using the two-tailed Student's *t*-test for normally distributed data with comparable variances. The Kolmogorov–Smirnov test was used for comparisons of two cumulative curves. Data sets containing more than two groups were tested using analyses of variance (ANOVA) and Bonferroni or Dunnett's *post hoc* test. The nonparametric Mann–Whitney *U*-test was used for data that were not normally distributed or when a normality test could not be applied. Statistical differences in cumulative survival were determined using the log-rank test. $P < 0.05$ was considered statistically significant.

**Expanded View** for this article is available online.

### Acknowledgements

We thank H. Okazawa and C. Yoshida for technical advice and helpful discussions on the histological analysis of motor neurons and R. F. Whittier and R. Ueta for critical reading of the manuscript and thoughtful discussions. We also thank J. Wilson for providing the AAV packaging plasmid (pRep2Cap9) and M. Nojima for advice on statistical analyses. This work was supported by Grant-in-Aid for JSPS Fellows Grant Number JP268885 (to S.M.), Grant-in-Aid of the Translational Research Network Program (B-15) from the Ministry of Education, Culture, Sports, Science and Technology of Japan (to Y.Y.), Grant-in-Aid for Scientific Research on Innovative Areas Grant Number JP25110711 (to Y.Y.), and the Practical Research Project for Rare/Intractable Diseases from Japan Agency for Medical Research and Development Grant Numbers 16ek0109003h0103 (to T.O.) and 16ek0109003h0003 (to Y.Y.).

### Author contributions

SM, TTe, SA, and YY designed research. SM and SA performed research. SM, TTo, and TO contributed to AAV production. SM, TTe, SA, TO, and YY analyzed data. SM, TTe, and YY wrote the manuscript.

---

**The paper explained**

**Problem**

Amyotrophic lateral sclerosis (ALS) is a progressive, multifactorial degenerative disease of motor neurons with severe muscle atrophy. The glutamate release inhibitor riluzole is the only medication approved by the FDA for ALS, but its therapeutic effects are limited, testifying to the strong need for new treatment strategies. The neuromuscular junction (NMJ), the essential synapse between a motor neuron and skeletal muscle, has recently emerged as an attractive therapeutic target, because studies of ALS model mice and patients revealed that degeneration of motor nerve terminals such as size reduction and denervation at NMJs precedes proximal motor neuron degeneration. However, NMJ-targeted therapies for ALS are yet to be developed.

**Results**

Therapeutic administration of an adeno-associated virus vector encoding *DOK7*, an essential gene for NMJ formation, suppressed size reduction of the motor nerve terminal and subsequent denervation at NMJs in *SOD1-G93A* ALS model mice (ALS mice). These findings demonstrate that *DOK7* gene therapy, which enlarges NMJs, has a protective effect against nerve terminal degeneration. Furthermore, the NMJ-targeted gene therapy suppressed muscle atrophy with no adverse effects on progressive proximal motor neuron death, and enhanced motor activity and life span in ALS mice.

**Impact**

This study establishes proof of concept that *DOK7* gene therapy, or potentially other methods that are able to enlarge NMJs after ALS onset, may be a novel treatment approach, either as a self-contained therapy or in combination with other therapies such as those aimed at promoting motor neuron survival. In addition, this therapeutic approach might be useful in other types of motor neuron diseases together with age-related muscle weakness, or sarcopenia, because degeneration at NMJs has also been reported in these disorders.

## Conflict of interest

The authors declare that they have no conflict of interest.

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
