## [Review Process File · EMBO Molecular Medicine]

DOK7 gene therapy enhances motor activity and life span in ALS model mice

Sadanori Miyoshi, Tohru Tezuka, Sumimasa Arimura, Taro Tomono, Takashi Okada & Yuji Yamanashi

Corresponding author: Yuji Yamanashi, The Institute of Medical Science, The University of Tokyo

Review timeline:

Submission date:	06 November 2016
Editorial Decision:	09 December 2016
Revision received:	08 March 2017
Editorial Decision:	03 April 2017
Revision received:	08 April 2017
Accepted:	12 April 2017

Transaction Report:

Editor: Céline Carret

1st Editorial Decision

09 December 2016

Thank you for the submission of your manuscript to EMBO Molecular Medicine. We have now heard back from the three referees whom we asked to evaluate your manuscript.

You will see that the three referees found the study clearly laid out, interesting, well performed and overall convincing. Nevertheless, suggestions are provided to make it even more compelling and I would like to encourage you to follow these lines during the revision of your article.

Please note that it is EMBO Molecular Medicine policy to allow only a single round of revision and that, as acceptance or rejection of the manuscript will depend on another round of review, your responses should be as complete as possible.

I look forward to receiving your revised manuscript.

***** Reviewer's comments *****

Referee #1 (Comments on Novelty/Model System):

The SOD1 ALS model is suitable to address gene therapy targeting NMJ. The role of the NMJ in ALS has been substantially addressed before. However, this study shows significant therapeutic efficacy of Dok7 overexpression using AAV-D7.

Referee #1 (Remarks):

Review of EMM-2016-07298

This article by S. Miyoshi and colleagues reports the use of an AAV-D7 vector to overexpress DOK7, an activator of the muscle-specific kinase MuSK, and stabilize neuromuscular junctions in ALS SOD1 G93A mice. Following systemic administration of AAV-D7 by intravenous injection at an early symptomatic stage, a significant protection of the neuromuscular junctions and skeletal muscle is observed, leading to improved motor performance and prolonged survival. In line with previous studies by the same group in the context of myopathies, these results support the application of this molecular approach to maintain the neuromuscular junction in ALS. Overall, it remains however unclear if AAV-D7 has modifying effects on disease progression. It is likely that the approach transiently stabilizes the remaining neuromuscular junction towards disease end stage.

Overall, the study is straightforward and well executed, and the manuscript well written. The reported data support the main conclusions of the study.

The following comments should be addressed before publication:

- Is there any evidence for a dysregulation of Dok7-MuSK in the muscle of the SOD1 G93A mice? If yes, is this rescued by AAV-D7? Of note, a recent study has reported impaired MuSK clustering in myofibers from SOD1 G93A mice (see Vilmont et al, 2016). The authors should at least discuss this observation with respect to their approach.
- In general authors should pay more attention to clearly indicate what is the muscle used for each Figure (diaphragm versus tibialis anterior). Along the same line, it would be useful to assess the neuromuscular junction occupancy in the same muscle group as the one used to assess protection of the muscle fibers. This is highly recommended as they authors stress the notion that the loss of neuromuscular junctions is the cause of muscle atrophy (see abstract).
- What is the proportion of the neuromuscular junctions that are partially occupied by nerve terminals? This question should be addressed as an increase in the size of the neuromuscular junction is shown here.
- Data should be provided regarding the pattern of transduction achieved following intravenous injection of AAV-D7. Panel 1b shows expression of EGFP in the diaphragm. The authors should provide other indications regarding the level of transduction in other groups of muscles. In addition, intravenous injection of AAV9 has been previously reported to lead to significant gene delivery in the central nervous system, both in the brain and spinal cord, and the vector was shown to preferentially target astrocytes (Foust KD et al, Nat Biotech 2009). Is EGFP expression found in the central nervous system in this experiment? If yes, in which cell types? This question is important, as the authors cannot exclude possible effects of AAV-D7 in tissues other than skeletal muscle. The authors should consider local injections in the skeletal muscle to verify that the observed effects solely result from the protection of neuromuscular junctions via transgene expression in the muscle tissue.

Minor comments:

- Descriptions of vector administration and injection method are missing.
- In the graphs showing $\Delta C(t)$ between the SOD1 transgene copies and the reference apob gene (panel 3e and panel 6e), it should be made more clear how is the subtraction calculated (human SOD1 transgene Ct - apob Ct).
- Figure 2, panel. The muscle fibers do not appear atrophied in the representative image provided for

ALS-NT.

- The ALS model used in this study (SOD1 G93A) should be indicated in the abstract.
- It is unclear what are the ALS control mice (ALS NT) used in the initial set of experiments? Did these mice receive a sham injection? The authors should clarify this. Ideally, a Table recapitulating the main values for the ALS NT/ALS D7 experiment and for the ALS EGFP/ALS NT experiment should be provided to facilitate the comparison between these two seemingly independent studies. I would also suggest that the authors indicate the change in survival time in this Table, as it is not obvious to find this value in the Figure legend.

Referee #2 (Remarks):

The manuscript by Miyoshi et al, shows that DOK7 gene therapy is beneficial in a preclinical model of ALS.

The work is clearly interesting, not only as it identifies DOK7 as a target for ALS therapies, but also as it focus on its role in the neuromuscular junctions (NMJs). Their major point is that strengthening the NMJ (via DOK7 gene therapy) has beneficial effects on the motor function of an ALS mouse model. The gene therapy seems to work distally at the NMJ, protecting the motor neurons from retracting from the NMJ, whilst having no effect in the neurodegeneration of the motor neuron soma (in the spinal cord).

There are a number of issues that will need to be addresses before publication, particularly:

1. One of the main conclusions of the manuscript is that DOK7 gene therapy protects NMJs, but has no effect on central motor neuron degeneration. This is based on assessing NMJs from the diaphragm at p120, and motor neurons from the lumbar spinal cord also at p120. However, the mice in the study survive to ~p150. Motor neurons in the spinal cord should be counted at end-stage (p150) to make sure that the protective effect is only happening distally at the NMJ. Moreover, the method for counting motor neurons should be explained more thoroughly in the materials. Counting motor neurons by Nissl staining is a commonly used method, but relies on the clear identification of motor neurons, how did the authors identified the motor neurons? Moreover, the authors are cutting thin sections (5micras) which is likely to lead to counting the same motor neuron more than once. Have they validated their counting method before? Ideally, the method should be validated by comparing the results obtained with the control group (G93A, non-treated) with the published literature.
2. Although mentioned throughout the manuscript, and also used to define disease onset, grip-strength data comparing the treated with non-treated groups after onset is not part of any figure. Why?
3. The notion of ALS being a distal neuropathy is controversial, but there are a number of examples in the literature that were missing from the discussion. For example, the work by Gould et al (*J. Neurosc.* 2006 Aug 23;26(34):8774-86) should be discussed.
4. Although not required for the present manuscript, to strengthen the notion that the protective effect is happening only distally, it would have been useful to assess the NMJs and estimate motor unit numbers at a hind-limb muscle like the EDL.

Referee #3 (Remarks):

This manuscript by Miyoshi et al. investigates the therapeutic effect of human DOK7 in ALS mice delivered via adeno-associated virus vectors. Previous study from the same group used similar approach to overexpress Dok7 gene, a muscle specific protein which is essential for neuromuscular junction formation, in a mouse model of myasthenia and Emery Dreifuss dystrophy. In this manuscript, the authors find that AAV-DOK7 treatment enhanced MuSK activation and increased area of motor nerve terminals, indicating a protection of Dok7 from nerve terminal degeneration in SOD1 mice. Although, the authors found that Dok7 gene therapy post-onset prolonged total life span by 12 days and duration after disease onset, there is no effect on motor neuron survival and the mechanism of action is not explored. The manuscript is straightforward, clearly laid out and includes important results, which reveal the therapeutic potential of targeting the neuro-muscular junction defects in ALS, even after disease onset. However, there are several points the authors should address to improve their work.

Major comments:

1. Results/methods (Figure 4): it should be noted that the open field measurement is an *in vivo* assay that can be beneficial for assessing activity levels that generally reflect locomotive function such as anxiety/depressive behaviors. It is a different measure than muscle strength i.e. grip strength, rotarod performance, clinical scoring. It usually makes a secondary or auxiliary outcome, accompanied by muscle strength measurements. Moreover, behavioral and locomotive activity is also influenced by additional factors (i.e. experimenter handling, environmental conditions, and cognition) creating considerable variation. Did the authors compare neurological score and grip strength between treated vs. untreated groups?
2. The Authors should describe how they distinguish motor neurons from other cells such as glial cells, oligos or dead/debris as they only used Nissl stain for this. They should insert a reference.
3. Supplementary fig 3: Is there significance between the treatments? Otherwise authors should rephrase the title. Do authors compare groups to the WT-NT or that of all groups?
4. Supplementary fig 5: Smaller Axonal diameters (ranging 3um-10um) in ALS mice have higher % of total axons than WT controls irrespective of treatment. This trend is reversed once axonal diameters "enlarge" at 9um-16um. As axonal diameter increases the % of total axons decrease - is there cell loss? Staining in panel A - authors should show a higher magnification image of this to show a clearer view of what the authors measure. Panel A) AAV-D7 the representative image looks like cell loss, did the authors account for this? Or make an observation using counter stain? Authors should label the panel with what they are staining with

Minor comments:

1. DOK7 gene therapy "enlarges the post synaptic AChR clusters size in ALS mice ALS untreated mice have equal levels to that of WT mice (Supplementary Fig 2). What is the significance of this? This should be discussed.
2. It would be nice to see electrophysiological effects of DOK7 gene therapy. Are these enhanced NMJs now "super synapses" and can therefore compensate for ongoing neuronal degeneration?
3. Authors do not discuss their results enough. For example: what are the consequences of the continued motor neuron death in the long-term? Can therapy help if motor neurons are still dying?
4. Figure 1: Where the images in b) imaged with the same settings/exposure times?
5. Results (figure 1): add reference and justify why they chose to assess the effect of AAV-DOK7 treatment on diaphragm muscle motor nerve terminals instead of limb terminals (PMID: 24040091)
6. Results (figure 2) "distribution of the fiber cross-section areas (CSAs) in tibialis anterior muscle was shifted upward by AAV-D7 treatment". This terminology is unclear. The authors should better describe the changes observed in myofiber composition and distribution
7. Figure 2: Insert high mag of myofiber for panel A. Authors should include pathological changes in muscle structure characteristic for ALS such as structure of the myofiber, cell infiltration etc. The authors can get a lot of important information from histology of muscle fibers.
8. There are some minor grammar mistakes throughout the manuscript
9. Typo: 42-43: mouse model and not model mice

Referee #1 (Comments on Novelty/Model System):

The SOD1 ALS model is suitable to address gene therapy targeting NMJ. The role of the NMJ in ALS has been substantially addressed before. However, this study shows significant therapeutic efficacy of Dok7 overexpression using AAV-D7.

Referee #1 (Remarks):

Review of EMM-2016-07298

This article by S. Miyoshi and colleagues reports the use of an AAV-D7 vector to overexpress DOK7, an activator of the muscle-specific kinase MuSK, and stabilize neuromuscular junctions in ALS SOD1 G93A mice. Following systemic administration of AAV-D7 by intravenous injection at an early symptomatic stage, a significant protection of the neuromuscular junctions and skeletal muscle is observed, leading to improved motor performance and prolonged survival. In line with previous studies by the same group in the context of myopathies, these results support the application of this molecular approach to maintain the neuromuscular junction in ALS.

Overall, it remains however unclear if AAV-D7 has modifying effects on disease progression. It is likely that the approach transiently stabilizes the remaining neuromuscular junction towards disease end stage.

Overall, the study is straightforward and well executed, and the manuscript well written. The reported data support the main conclusions of the study.

The following comments should be addressed before publication:

Is there any evidence for a dysregulation of Dok7-MuSK in the muscle of the SOD1 G93A mice? If yes, is this rescued by AAV-D7? Of note, a recent study has reported impaired MuSK clustering in myofibers from SOD1 G93A mice (see Vilmont et al, 2016). The authors should at least discuss this observation with respect to their approach

Dok-7-MuSK is indispensable for the formation and maintenance of the postsynaptic apparatus of the NMJ, and impairment of this signaling leads to size-reduction of the postsynaptic structure. However, the size of the postsynaptic area was not affected in the diaphragm muscle of ALS mice at the P120 symptomatic stage (Fig 1E, in the revised manuscript), indicating that Dok-7-MuSK is within the safety margin of the homeostasis of NMJs. Nevertheless, as the referee suggested, we added the statement to Discussion of the revised manuscript “mislocalization of MuSK was reported in *SOD1-G93A* ALS model mice” and cited the report by Vilmont et al.

In general authors should pay more attention to clearly indicate what is the muscle used for each Figure (diaphragm versus tibialis anterior). Along the same line, it would be useful to assess the neuromuscular junction occupancy in the same muscle group as the one used to assess protection of the muscle fibers. This is highly recommended as they authors stress the notion that the loss of neuromuscular junctions is the cause of muscle atrophy (see abstract).

We agree with the referee that we should pay more attention to clearly show which muscle was used for each figure. Although we indicated it in the main text, figure legends and Materials and Methods for almost all figures in the original manuscript, we failed to do it for Fig. 1C in the main text. Therefore, we added the statement “in the hind-limb muscle” to the corresponding description of Fig. 1C in Results of the revised manuscript.

With regard to the referee’s recommendation that we assess the NMJ size in the same muscle group as was used to assess protection of the muscle fiber, we failed to obtain reliable data on NMJ size in the latter muscle (tibialis anterior) of ALS mice. Therefore, we used the diaphragm muscle, where NMJs are particularly amenable to whole-mount imaging due to the muscle’s thin and planar structure (Tetruashvily et al, 2016) and where changes in neuromuscular transmission occur long before motor symptom onset (Rocha et al, 2013). To

make it clear why we chose the diaphragm muscle to assess NMJs, we added this information to Results of the revised manuscript. Please note that Referee #3 requested addition of the reference (Rocha *et al*, 2013) and a related rationale to justify why we chose to assess the effect of AAV-DOK7 treatment on diaphragm muscle motor nerve terminals instead of limb terminals (please see below). Furthermore, as the referee pointed out in this comment, we added the statement “to clarify the therapeutic effects of NMJ protection on muscle weakness, electrophysiological study of affected muscle will be essential” to Discussion of the revised manuscript. Please note that Referee #2 also mentioned “Although not required for the present manuscript, to strengthen the notion that the protective effect is happening only distally, it would have been useful to assess the NMJs and estimate motor unit numbers at a hind-limb muscle like the EDL” (please see below).

What is the proportion of the neuromuscular junctions that are partially occupied by nerve terminals? This question should be addressed as an increase in the size of the neuromuscular junction is shown here.

In accord with the referee’s comment, we stated in Results of the revised manuscript “Indeed, denervation at NMJs was significantly suppressed by the treatment (Fig 1G), although these NMJs showed partial innervation partly due to the primary enlargement of the postsynaptic area (Fig 1D-F)”. In addition, to maintain the logical flow after this addition, we added a small change to the next sentence. Please note that, even in wild-type mice, NMJs showed partial innervation due to the primary enlargement of the postsynaptic area by AAV-D7 treatment (Arimura *et al*, 2014).

*Data should be provided regarding the pattern of transduction achieved following intravenous injection of AAV-D7. Panel 1b shows expression of EGFP in the diaphragm. The authors should provide other indications regarding the level of transduction in other groups of muscles. In addition, intravenous injection of AAV9 has been previously reported to lead to significant gene delivery in the central nervous system, both in the brain and spinal cord, and the vector was shown to preferentially target astrocytes (Foust KD *et al*, Nat Biotech 2009). Is EGFP expression found in the central nervous system in this experiment? If yes, in which cell types? This question is important, as the authors cannot exclude possible effects of AAV-D7 in tissues other than skeletal muscle. The authors should consider local injections in the skeletal muscle to verify that the observed effects solely result from the protection of neuromuscular junctions via transgene expression in the muscle tissue.*

In accord with the referee’s comment, we examined Dok-7-EGFP transduction in several muscle groups (tibialis anterior muscle, extensor digitorum longus muscle, soleus muscle and diaphragm muscle) along with the spinal cord and brain as additional experiments. These data were included in the revised manuscript as Fig. EV1. Please note that EGFP-fluorescence was measured in the same experimental settings in this assay as described in Materials and Methods of the revised manuscript. Although the nature of EGFP-positive cells are yet to be determined, weak or faint EGFP signals in the spinal cord and the cerebellum of brain were observed (Fig EV1B and C). Thus, in accord with the referee’s suggestion, we added the statement “Given that EGFP-fluorescence from AAV-D7 infected cells was detectable however weakly and faintly in the spinal cord and cerebellum, respectively, in AAV-D7-treated mice (Fig EV1B and C), skeletal muscle-specific transduction of *DOK7* will be necessary to completely exclude the possible contribution of non-muscle transgene expression” to Discussion of the revised manuscript.

Minor comments:

Descriptions of vector administration and injection method are missing.

With regard to this point, we added the description of vector administration and the injection method to Material and Methods of the revised manuscript as follows:

“In vivo AAV injection

1.2×10^{12} vg of AAV-D7 or AAV-EGFP was intravenously injected by a single shot via the tail vein at a symptomatic stage (P90) or individually defined disease onset. Note that we used

wild-type or ALS mice without any treatment, even sham injection, as the control “non-treated mice” or “WT - NT” or “ALS - NT”

In the graphs showing $\Delta C(t)$ between the SOD1 transgene copies and the reference apob gene (panel 3e and panel 6e), it should be made more clear how is the subtraction calculated (human SOD1 transgene Ct - apob Ct).

In accord with the referee’s comment, we added the statement “To calculate the human transgene level, the ΔC_t value of hSOD1 was subtracted from the ΔC_t value of apob” in the corresponding figure legends (Fig 1B, Fig 3E, Fig EV3E) of the revised manuscript.

Figure 2, panel. The muscle fibers do not appear atrophied in the representative image provided for ALS-NT.

In accord with the referee’s comment, we revised the corresponding images in Fig. 2A with the addition of magnified images (Fig 2B) in the revised manuscript, where sites of severe myofiber atrophy were indicated by arrowheads.

The ALS model used in this study (SOD1 G93A) should be indicated in the abstract.

In accord with the referee’s comment, we added the statement “in the SOD1-G93A ALS mouse model” to the Abstract of the revised manuscript.

It is unclear what are the ALS control mice (ALS NT) used in the initial set of experiments? Did these mice receive a sham injection? The authors should clarify this. Ideally, a Table recapitulating the main values for the ALS NT/ALS D7 experiment and for the ALS EGFP/ALS NT experiment should be provided to facilitate the comparison between these two seemingly independent studies. I would also suggest that the authors indicate the change in survival time in this Table, as it is not obvious to find this value in the Figure legend.

We agree with the referee that we should clarify the nature of “ALS NT” and also “WT-NT”. Thus, we added the statement “Note that we used wild-type or ALS mice without any treatment, even sham injection, as the control “non-treated mice” or “WT - NT” or “ALS - NT”” to Materials and Methods of the revised manuscript. In addition, in accord with the referee’s suggestion, we added this description of the survival time to Results in the revised manuscript: “DOK7 gene therapy significantly prolonged life span of ALS mice (166.3 ± 3.2 days) compared with non-treated ALS mice (154.4 ± 2.7 days) (Fig 3A). This therapy also prolonged duration of survival after ALS onset (64.2 ± 3.3 days) compared with non-treated mice (50.3 ± 3.1 days) (Fig 3B)” and “treatment with AAV-EGFP did not significantly alter either the survival (162.1 ± 2.5 days) nor duration after onset (59.8 ± 3.0 days) compared with non-treated mice (survival: 158.4 ± 1.9 days, duration after onset: 57.3 ± 2.4 days)”.

Moreover, with the referee’s comment “the comparison between these two seemingly independent studies”, we realized that it would be better for the study to make them (the ALS NT/ALS D7 and the ALS EGFP/ALS NT experiments) independent. Thus, we additionally performed the latter experiment independently of the former and added the new data as Fig. EV3 in the revised manuscript.

Referee #2 (Remarks):

The manuscript by Miyoshi et al, shows that DOK7 gene therapy is beneficial in a preclinical model of ALS.

The work is clearly interesting, not only as it identifies DOK7 as a target for ALS therapies, but also as it focus on its role in the neuromuscular junctions (NMJs). Their major point is that strengthening the NMJ (via DOK7 gene therapy) has beneficial effects on the motor function of an ALS mouse model. The gene therapy seems to work distally at the NMJ, protecting the motor neurons from retracting from the NMJ, whilst having no effect in the neurodegeneration of the motor neuron soma (in the spinal cord).

There are a number of issues that will need to be addressed before publication, particularly:

1. One of the main conclusions of the manuscript is that DOK7 gene therapy protects NMJs, but has no effect on central motor neuron degeneration. This is based on assessing NMJs from the diaphragm at p120, and motor neurons from the lumbar spinal cord also at p120. However, the mice in the study survive to ~p150. Motor neurons in the spinal cord should be counted at end-stage (p150) to make sure that the protective effect is only happening distally at the NMJ. Moreover, the method for counting motor neurons should be explained more thoroughly in the materials. Counting motor neurons by Nissl staining is a commonly used method, but relies on the clear identification of motor neurons, how did the authors identify the motor neurons? Moreover, the authors are cutting thin sections (5micras) which is likely to lead to counting the same motor neuron more than once. Have they validated their counting method before? Ideally, the method should be validated by comparing the results obtained with the control group (G93A, non-treated) with the published literature.

We agree with the referee that motor neurons in the spinal cord should be counted at end-stage (p150) to make sure that the protective effect is only happening distally at the NMJ. Thus, we performed additional experiments and counted the number of motor neuron cell bodies at P150 in ALS mice treated or not with AAV-D7 after individually defined disease onset. We added these data as Fig. EV4A and EV4B in the revised manuscript, demonstrating the number of motor neuron cell bodies was unchanged even at end stage by DOK7 gene therapy. Therefore, we added the statement “Note that DOK7 gene therapy did not affect degeneration of motor neuron cell bodies even at end stage (P150) in ALS mice (Fig EV4A and B), implying that the protective effect of this therapy may be confined distally at the NMJ” to Results of the revised manuscript.

With regard to the technical concern on Nissl staining, we agree with the referee that it would be better to make a more detailed description of the method. Thus, we add the following statement with an additional reference to Materials and Methods of the revised manuscript: “The following criteria were used to count motor neurons: (1) cells located in the ventral horn and (2) cells with a maximum diameter of 20 μm or more (Cai *et al*, 2015)”. In addition, we also agree with the referee that we might have counted a single motor neuron more than once in the assay in Fig. EV2B of the revised manuscript. Thus, in accord with the referee’s suggestion, we validated the method by comparing our data with those in the previous study (Yoo & Ko, 2012), which used 25-μm sections instead of 5-μm. To include this information, we added the statement to Materials and Methods of the revised manuscript “Note that we validated this method by comparing the motor neuron number in wild-type or ALS mice at p120 (Fig EV2B) with those reported elsewhere (Yoo & Ko, 2012)”.

2. Although mentioned throughout the manuscript, and also used to define disease onset, grip-strength data comparing the treated with non-treated groups after onset is not part of any figure. Why?

As the referee pointed out, we had performed grip strength test up to 152 days of age, which failed to show significant benefits of DOK7 gene therapy. However, with the referee’s concern and also with Referee #3’s comment (please see below), we reconsidered and included the data as Fig. EV4C in the revised manuscript and added the statement “Although we monitored the grip strength of individual mice up to 152 days of age, the data fail to show a significant difference between AAV-D7-treated and non-treated ALS mice (Fig EV4C)” to Results.

*3. The notion of ALS being a distal neuropathy is controversial, but there are a number of examples in the literature that were missing from the discussion. For example, the work by Gould *et al* (J. Neurosc. 2006 Aug 23;26(34):8774-86) should be discussed.*

We agree with the referee that we should discuss related works including one by Gould *et al*. Thus, we added the statement to Discussion of the revised manuscript “previous studies have shown that degeneration of motor nerve terminals at NMJs occurs independently of motor neuron cell death in ALS model mice (Gould *et al*, 2006; Kostic *et al*, 1997; Parone *et al*, 2013)”, which nicely provides a rationale for the following, original discussion “these NMJ-

targeting therapies might be more effective when used in combination with other therapies such as those aimed at promoting motor neuron survival”.

4. Although not required for the present manuscript, to strengthen the notion that the protective effect is happening only distally, it would have been useful to assess the NMJs and estimate motor unit numbers at a hind-limb muscle like the EDL.

We agree with the referee’s comment that, for future study, assessment of NMJs and estimation of motor unit numbers at a hind-limb muscle would be useful to strengthen the notion that the protective effect is happening only distally. Thus, we added a statement to Discussion of the revised manuscript “In addition, to clarify the therapeutic effects of NMJ protection on muscle weakness, electrophysiological study of affected muscle will be essential”. Moreover, with a related suggestion from Referee #1, we performed additional experiments and counted the number of motor neuron cell bodies at end stage (P150) in ALS mice treated or not with AAV-D7 after individually defined disease onset. We added the statement about the data “Note that *DOK7* gene therapy did not affect degeneration of motor neuron cell bodies even at end stage (P150) in ALS mice (Fig EV4A and B), implying that the protective effect of this therapy may be confined distally at the NMJ” to Results of the revised manuscript.

Referee #3 (Remarks):

*This manuscript by Miyoshi et al. investigates the therapeutic effect of human *DOK7* in ALS mice delivered via adeno-associated virus vectors. Previous study from the same group used similar approach to overexpress *Dok7* gene, a muscle specific protein which is essential for neuromuscular junction formation, in a mouse model of myasthenia and Emery Dreifuss dystrophy. In this manuscript, the authors find that AAV-*DOK7* treatment enhanced MuSK activation and increased area of motor nerve terminals, indicating a protection of *Dok7* from nerve terminal degeneration in *SOD1* mice. Although, the authors found that *Dok7* gene therapy post-onset prolonged total life span by 12 days and duration after disease onset, there is no effect on motor neuron survival and the mechanism of action is not explored. The manuscript is straightforward, clearly laid out and includes important results, which reveal the therapeutic potential of targeting the neuro-muscular junction defects in ALS, even after disease onset. However, there are several points the authors should address to improve their work.*

Major comments:

1. Results/methods (Figure 4): it should be noted that the open field measurement is an in vivo assay that can be beneficial for assessing activity levels that generally reflect locomotive function such as anxiety/depressive behaviors. It is a different measure than muscle strength i.e. grip strength, rotarod performance, clinical scoring. It usually makes a secondary or auxiliary outcome, accompanied by muscle strength measurements. Moreover, behavioral and locomotive activity is also influenced by additional factors (i.e. experimenter handling, environmental conditions, and cognition) creating considerable variation. Did the authors compare neurological score and grip strength between treated vs. untreated groups?

We agree with the referee’s comment that open field measurement reflects not only motor function but also anxiety/depressive behavior and is influenced by additional factors. Although these indirect elements may also affect measurements of muscle strength in general, it would be better for the study to perform more direct measurements. Indeed, as mentioned in the original manuscript, we performed grip strength tests up to 152 days of age but failed to show significant benefits of *DOK7* gene therapy. Nonetheless, with this comment and also with Referee #2’s concern (please see above), we reconsidered and included the data as Fig. EV4C in the revised manuscript and added the statement “Although we monitored the grip strength of individual mice up to 152 days of age, the data fail to show a significant difference between AAV-D7-treated and non-treated ALS mice (Fig EV4C)” to Results. This should help readers correctly understand that *DOK7* gene therapy enhanced locomotor activity but failed to significantly improve grip strength in ALS mice. Just to be clear, we did not evaluate neurological scores of mice in the study.

2. The Authors should describe how they distinguish motor neurons from other cells such as glial cells, oligos or dead/debris as they only used Nissl stain for this. They should insert a reference.

Based on essentially the same concern, Referee #2 also suggested that we describe how motor neurons were counted and further validate the method by comparing our results with those in the published literature (please see above). Thus, we added the statement with an additional reference, as this referee (#3) suggested, to Materials and Methods of the revise manuscript "The following criteria were used to count motor neurons: (1) cells located in the ventral horn and (2) cells with a maximum diameter of 20 μm or more (Cai *et al*, 2015)". In addition, we validated the method by comparing our data with those in the previous study (Yoo & Ko, 2012), which used 25- μm sections instead of 5- μm . Thus, in the revised manuscript, we added the statement to Materials and Methods "Note that we validated this method by comparing the motor neuron number in wild-type or ALS mice at p120 (Fig EV2B) with those reported elsewhere (Yoo & Ko, 2012)".

3. Supplementary fig 3: Is there significance between the treatments? Otherwise authors should rephrase the title. Do authors compare groups to the WT-NT or that of all groups?

We agree with the referee that the title of Supplementary fig. 3 in the original manuscript is misleading, because there is no significant change supporting it. However, in the revised manuscript, we moved the data into Fig. 2 (as Fig 2C), in which Fig. 2D and 2E support Fig. 2's title "*DOK7* gene therapy suppresses myofiber atrophy in ALS mice".

4. Supplementary fig 5: Smaller Axonal diameters (ranging 3um-10um) in ALS mice have higher % of total axons than WT controls irrespective of treatment. This trend is reversed once axonal diameters "enlarge" at 9um-16um. As axonal diameter increases the % of total axons decrease - is there cell loss? Staining in panel A - authors should show a higher magnification image of this to show a clearer view of what the authors measure. Panel A) AAV-D7 the representative image looks like cell loss, did the authors account for this? Or make an observation using counter stain? Authors should label the panel with what they are staining with

As shown in Supplementary fig. 4 in the original manuscript, the number of motor neuron cell bodies was decreased in ALS mice at P120 irrespectively of AAV-D7 treatment as compared to that in wild-type mice, indicating cell loss in ALS mice. Given the referee's comment, we combined Supplementary fig. 4 and 5 in the original manuscript into Fig. EV2 in the revised manuscript. This should help readers comprehensively understand that *DOK7* gene therapy shows no obvious effects on motor neuron survival and motor axon atrophy at P120 in ALS mice.

Moreover, in accord with the referee's suggestion, we included higher magnification insets into Fig. EV2C and labeled the panel "Neurofilament-H".

Minor comments:

1. *DOK7* gene therapy "enlarges the post synaptic AChR clusters size in ALS mice ALS untreated mice have equal levels to that of WT mice (Supplementary Fig 2). What is the significance of this? This should be discussed.

As shown in Supplementary fig. 2 in the original manuscript and in Fig. 1E in the revised manuscript, there is no significant difference in AChR cluster size between wild-type and ALS mice at P120 in the absence of AAV-D7 treatment. In accord with the referee's comment, we added the discussion "In non-treated ALS mice, although the postsynaptic area characterized by clustered AChRs was not affected, the area of motor nerve terminals was significantly decreased, supporting the neuropathic nature of the defects" to Results of the revised manuscript. This should help readers understand the implication of the data.

2. It would be nice to see electrophysiological effects of *DOK7* gene therapy. Are these enhanced NMJs now "super synapses" and can therefore compensate for ongoing neuronal degeneration?

In accord with this comment and related suggestions from Referees #1 and #2 (please see above), we added the statement “In addition, to clarify the therapeutic effects of NMJ protection on muscle weakness, electrophysiological study of affected muscle will be essential” to Discussion of the revised manuscript. We would like to tackle this point in future works.

3. Authors do not discuss their results enough. For example: what are the consequences of the continued motor neuron death in the long-term? Can therapy help if motor neurons are still dying?

In accord with Referee #2’s suggestion (please see above), we additionally counted the number of motor neuron cell bodies at P150 in ALS mice treated or not with AAV-D7 after individually defined disease onset, and added the statement “Note that *DOK7* gene therapy did not affect degeneration of motor neuron cell bodies even at end stage (P150) (Fig EV4A and B), implying that the protective effect of this therapy may be confined distally at the NMJ” to Results of the revised manuscript. Also, in response to another comment from Referee #2, we added the statement to Discussion of the revised manuscript “previous studies have shown that degeneration of motor nerve terminals at NMJs occurs independently of motor neuron cell death in ALS model mice (Gould *et al*, 2006; Kostic *et al*, 1997; Parone *et al*, 2013)”, which provides a rationale for the following discussion: “these NMJ-targeting therapies might be more effective when used in combination with other therapies such as those aimed at promoting motor neuron survival”. We believe that these discussions also answer Referee #3’s questions.

4. Figure 1: Where the images in b) imaged with the same settings/exposure times?

Yes, they were imaged with the same settings/exposure times. Thus, to clarify this point, we added the statement “Images were captured with the same settings and exposure time in each experimental group for comparison” to Materials and Methods (Whole-mount staining of NMJs) in the revised manuscript.

5. Results (figure 1): add reference and justify why they chose to assess the effect of AAV-DOK7 treatment on diaphragm muscle motor nerve terminals instead of limb terminals (PMID: 24040091)

In accord with the referee’s comment including the suggested reference, we added the statement to Results of the revised manuscript “To examine histopathology, we first studied NMJs in the diaphragm muscle, where NMJs are particularly amenable to whole-mount imaging due to the muscle’s thin and planar structure (Tetrushvily *et al*, 2016) and where changes in neuromuscular transmission occur long before motor symptom onset (Rocha *et al*, 2013)”.

6. Results (figure 2) "distribution of the fiber cross-section areas (CSAs) in tibialis anterior muscle was shifted upward by AAV-D7 treatment". This terminology is unclear. The authors should better describe the changes observed in myofiber composition and distribution

This figure (Supplementary fig. 3) in the original manuscript was moved to Fig. 2C and, in accord with the referee’s comment, we modified the corresponding statement: “Analysis of size distribution of the fiber cross-section areas (CSA) in tibialis anterior muscle showed that AAV-D7 treatment apparently increased the proportion of myofibers with relatively larger diameter in ALS mice compared with non-treated ALS mice (Fig 2A-C)” to Results of the revised manuscript.

7. Figure 2: Insert high mag of myofiber for panel A. Authors should include pathological changes in muscle structure characteristic for ALS such as structure of the myofiber, cell infiltration etc. The authors can get a lot of important information from histology of muscle fibers.

In accord with the referee’s comment, we added high magnification images (Fig 2B) to the revised manuscript, in which we found and showed sites of severe myofiber atrophy in non-treated ALS mice.

8. There are some minor grammar mistakes throughout the manuscript

We corrected several grammatical errors found in the original manuscript.

9. Typo: 42-43: mouse model and not model mice

This inappropriate wording was deleted in editorial formatting of the revised manuscript.

References

- Arimura S, Okada T, Tezuka T, Chiyo T, Kasahara Y, Yoshimura T, Motomura M, Yoshida N, Beeson D & Yamanashi Y (2014) *DOK7* gene therapy benefits mouse models of diseases characterized by defects in the neuromuscular junction. *Science* 345: 1505–1508
- Cai MD, Choi SM & Yang EJ (2015) The effects of bee venom acupuncture on the central nervous system and muscle in an animal hSOD1^{G93A} mutant. *Toxins (Basel)*. 7: 846–858
- Gould TW, Buss RR, Vinsant S, Prevette D, Sun W, Knudson CM, Milligan CE & Oppenheim RW (2006) Complete dissociation of motor neuron death from motor dysfunction by Bax deletion in a mouse model of ALS. *J. Neurosci.* 26: 8774–8786
- Kostic V, Jackson-Lewis V, de Bilbao F, Dubois-Dauphin M & Przedborski S (1997) Bcl-2: prolonging life in a transgenic mouse model of familial amyotrophic lateral sclerosis. *Science* 277: 559–562
- Parone PA, Da Cruz S, Han JS, McAlonis-Downes M, Vetto AP, Lee SK, Tseng E & Cleveland DW (2013) Enhancing Mitochondrial Calcium Buffering Capacity Reduces Aggregation of Misfolded SOD1 and Motor Neuron Cell Death without Extending Survival in Mouse Models of Inherited Amyotrophic Lateral Sclerosis. *J. Neurosci.* 33: 4657–4671
- Rocha MC, Pousinha PA, Correia AM, Sebastião AM & Ribeiro JA (2013) Early Changes of Neuromuscular Transmission in the SOD1(G93A) Mice Model of ALS Start Long before Motor Symptoms Onset. *PLoS One* 8: e73846
- Tetruashvily MM, McDonald MA, Frieze KK & Boulanger LM (2016) MHCI promotes developmental synapse elimination and aging-related synapse loss at the vertebrate neuromuscular junction. *Brain. Behav. Immun.* 56: 197–208
- Vilmont V, Cadot B, Vezin E, Le Grand F & Gomes ER (2016) Dynein disruption perturbs post-synaptic components and contributes to impaired MuSK clustering at the NMJ: implication in ALS. *Sci. Rep.* 6: 27804
- Yoo YE & Ko CP (2012) Dihydrotestosterone ameliorates degeneration in muscle, axons and motoneurons and improves motor function in amyotrophic lateral sclerosis model mice. *PLoS One* 7: e37258

2nd Editorial Decision

03 April 2017

Thank you for the submission of your revised manuscript to EMBO Molecular Medicine. We have now received the enclosed reports from the referees that were asked to re-assess it. As you will see the reviewers are now globally supportive and I am pleased to inform you that we will be able to accept your manuscript pending the following final amendments:

- 1) Please address the minor comments of referee 1. Please provide a letter INCLUDING the reviewer's reports and your detailed responses to their comments (as Word file).
- 2) You have 4 figures and 4 EV figures. However, you have indicated that the study was a report. For it to be a report, it should have 3 figures. I would like to suggest that you move one of the main figure to an EV figure. Please make sure to amend all callouts and labels accordingly.

Please submit your revised manuscript within two weeks. I look forward to seeing a revised form of your manuscript as soon as possible.

***** Reviewer's comments *****

Referee #1 (Remarks):

The Authors have properly addressed most of the comments and the manuscript has been

substantially improved.

However, as a follow up on our initial comments, there are two points that could have been better addressed:

1- From the data provided by the Authors, it is increasingly evident that the AAV-D7 treatment preserves the neuromuscular junctions without protecting the motoneurons in the spinal cord. It is surprising that this aspect is only briefly addressed in the last paragraph of the Discussion. The manuscript could be improved by discussing the following points: is there a disease-resistant subtype of motoneurons that is more likely to reinnervate the neuromuscular junctions left unoccupied by the neurons that first degenerate? Is the reinnervation facilitated by AAV-D7 mainly because the motor endplates are enlarged?

2- The Authors have chosen not to assess the occupancy of the neuromuscular junctions in the tibialis anterior muscle as they failed to perform whole-mount imaging to visualise these structures. Although we agree that whole-mount imaging of hindlimb muscles is technically challenging, we would like to emphasise that the occupancy of the neuromuscular junctions could be instead quantified on 20µm sections of the muscle. On these sections, it is also possible to co-stain the pre- and post-synaptic compartments to verify partial or complete co-localization.

Referee #2 (Remarks):

After revisions, the manuscript is now acceptable for publication

Referee #3 (Remarks):

The authors addressed all comment and the manuscript is ready to be published

2nd Revision - authors' response

08 April 2017

Referee #1 (Remarks):

The Authors have properly addressed most of the comments and the manuscript has been substantially improved. However, as a follow up on our initial comments, there are two points that could have been better addressed:

We are pleased to know that our revisions are mostly satisfactory to this referee, and appreciate the helpful criticisms and suggestions including the following two points.

1- From the data provided by the Authors, it is increasingly evident that the AAV-D7 treatment preserves the neuromuscular junctions without protecting the motoneurons in the spinal cord. It is surprising that this aspect is only briefly addressed in the last paragraph of the Discussion. The manuscript could be improved by discussing the following points: is there a disease resistant subtype of motoneurons that is more likely to reinnervate the neuromuscular junctions left unoccupied by the neurons that first degenerate? Is the reinnervation facilitated by AAV-D7 mainly because the motor endplates are enlarged?

Although we mentioned in the manuscript that *DOK7* gene therapy might counteract size-reduction of the motor nerve terminals at NMJs in ALS mice, there remains the possibility that this therapy might also facilitate reinnervation by remaining axons as the referee kindly pointed out. Because we believe that in-depth discussion of this particular possibility is beyond the scope of the present study, we added the following statement to Results of the revised manuscript (p.8, l. 17): “This is apparently consistent with the reports that degeneration of motor nerve terminals at NMJs occurs independently of motor neuron cell death (Gould *et al*, 2006; Kostic *et al*, 1997; Parone *et al*, 2013). In addition, although we mentioned above that *DOK7* gene therapy might counteract size-reduction of the motor nerve terminals at NMJs in ALS mice, there remains the possibility that this therapy might also

facilitate reinnervation by remaining motor neurons. Thus, it is important to investigate how *DOK7* gene therapy inhibits degeneration of the motor nerve terminals in ALS mice". In addition, to maintain the logical flow after this addition, we added the statement "as mentioned above" to Discussion of the re-revised manuscript (p.12, l. 3).

2- The Authors have chosen not to assess the occupancy of the neuromuscular junctions in the tibialis anterior muscle as they failed to perform whole-mount imaging to visualise these structures. Although we agree that whole-mount imaging of hindlimb muscles is technically challenging, we would like to emphasise that the occupancy of the neuromuscular junctions could be instead quantified on 20µm sections of the muscle. On these sections, it is also possible to co-stain the pre- and post-synaptic compartments to verify partial or complete co-localization.

We appreciate this detailed technical advice for future study of the tibialis anterior muscle. In particular, we would like to see how it works on damaged muscle and NMJs in ALS mice, because we failed to obtain reliable data on NMJ size in the tibialis anterior muscle of ALS mice.

Referee #2 (Remarks):

After revisions, the manuscript is now acceptable for publication

We are pleased to know that our revisions are satisfactory to this referee, and appreciate the helpful criticisms and suggestions.

Referee #3 (Remarks):

The authors addressed all comment and the manuscript is ready to be published

We are pleased to know that our revisions are satisfactory to this referee, and appreciate the helpful criticisms and suggestions.

Editor (Remarks):

You have 4 figures and 4 EV figures. However, you have indicated that the study was a report. For it to be a report, it should have 3 figures.

We agree with the editor, and thus combined previous Fig. 3 and Fig. 4 to make new Fig. 3, which includes former Fig. 4A and 4B as new Fig. 3F and 3G, respectively, together with its new legend in the re-revised manuscript.

In the main article file, please do the following:

-remove the legend of Movie EV1

-remove text highlights

-in M&M, provide antibody dilutions that were used for each antibody

-indicate in legends exact p= values, not a range. Some people found that to keep the figures clear, providing a supplemental table with all exact p-values was preferable. You are welcome to do this if you want to.

-in M&M, Statistical paragraph, you indicated "data analysed using EZR or JMP" please define acronyms and/or softwares as appropriate.

-Call outs for EV1A are missing. Figure 4 and figure EV3 are call for as a whole, not by panels. Please correct.

We appreciate these detailed and helpful comments, and thus accordingly prepared the re-revised manuscript. Please note that we indicated exact *P*-values rounded off to four decimal place, but those less than 0.0001 are exceptionally indicated as "*P* < 0.0001" to avoid indicating those less than 0.00005 as "*P* = 0.0000" or "*P* = 0", given that this type of indication is also seen in several papers published recently in *EMBO Molecular Medicine*.

Corresponding Author Name: Yuji Yamanashi

Journal Submitted to: EMBO molecular medicine

Manuscript Number: EMM-2016-07298